# A Study on the Chemistry and Biological Activity of 26-Sulfur Analogs of Diosgenin: Synthesis of 26-Thiodiosgenin *S*-Mono- and Dioxides, and Their Alkyl Derivatives

**DOI:** 10.3390/molecules28010189

**Published:** 2022-12-26

**Authors:** Aneta M. Tomkiel, Dorota Czajkowska-Szczykowska, Ewa Olchowik-Grabarek, Lucie Rárová, Szymon Sękowski, Jacek W. Morzycki

**Affiliations:** 1Laboratory of Natural Products, Department of Organic Chemistry, Faculty of Chemistry, University of Bialystok, K. Ciołkowskiego 1K, 15-245 Białystok, Poland; 2Laboratory of Molecular Biophysics, Department of Microbiology and Biotechnology, Faculty of Biology, University of Bialystok, K. Ciołkowskiego 1 J, 15-245 Białystok, Poland; 3Department of Experimental Biology, Faculty of Science, Palacký University, Šlechtitelů 27, CZ-78371 Olomouc, Czech Republic

**Keywords:** 26-thiodiosgenin, thiol oxidation, sulfoxides, sulfones, sapogenins, steroids, antimicrobial activity, cytotoxicity

## Abstract

A chemoselective procedure for MCPBA oxidation of 26-thiodiosgenin to corresponding sulfoxides and sulfone was elaborated. An unusual equilibration of sulfoxides in solution was observed. Moreover, α-alkylation of sulfoxide and sulfone was investigated. Finally, the biological activity of obtained compounds was examined.

## 1. Introduction

Saponins are a class of chemical compounds abundant in various plant species [1]. More specifically, they are amphiphilic glycosides producing the soap-like foam when shaken in aqueous solutions. The lipophilic triterpene or steroid aglycones (sapogenins) are combined in these compounds with one or more hydrophilic sugar moieties. The steroidal sapogenins usually contain 27 carbon atoms and the oxidized side chain, which forms a spiroacetal system characteristic for steroidal spirostanes, e.g., diosgenin (**1**) (Figure 1). These compounds have received considerable attention as precursors for synthesizing sex hormones and various steroidal drugs.

There is also a group of naturally occurring compounds based on a C_27_ cholestane skeleton, which are essentially nitrogen analogs of spirostane sapogenins, e.g., solasodine (**2**) (Figure 1). These steroidal alkaloids (spirosolanes) are a class of secondary metabolites isolated from plants (mostly of the family *Solanaceae*), amphibians, and marine invertebrates [2,3,4,5]. Evidence accumulated in the last two decades demonstrates that steroidal alkaloids of the spirosolane group show a wide range of bioactivities, including anticancer, antimicrobial, anti-inflammatory, antinociceptive, etc., suggesting their great potential for pharmaceutical application. Several comprehensive review articles on the alkaloid bioactivity, especially anticancer activity, and the mechanism of their biological action have recently appeared [6,7,8].

The replacement of the F-ring oxygen atom in spirostane sapogenins with a different heteroatom severely affects the chemical properties of a steroid and may result in useful alterations to its biological activity. The potential of heterosteroids as novel drugs encourages organic chemists to undertake studies in this field. The sulfur analogs of steroidal sapogenins do not occur in nature but 26-thiodiosgenin (**3**), the diosgenin F-ring thia-counterpart, was described in the literature many years ago [9]. However, efficient syntheses of this compound have been reported only recently. In particular, the one-pot Wang synthesis [10], which involves the treatment of a solution of diosgenin in dichloromethane with hydrogen sulfide gas in the presence of BF_3_·Et_2_O as a catalyst, was the most advantageous for us. It has been shown that replacing the F-ring oxygen atom of diosgenin with sulfur increases compound cytotoxicity against different cancer cell lines, especially in the case of glycosyl derivatives. For example, IC_50_ of a natural saponin, dioscine, against the lung cancer cell line A549 IC_50_ was found 4.02 μM versus 3.72 μM for 26-thiodioscine [11,12]. Though 26-thiodiosgenin (**3**) is now readily available from diosgenin, its chemistry has not been explored yet. The *Se*-analog of diosgenin (**4**), in which selenium atom replaces the F-ring spiroketal oxygen, has also been described, but we have found the literature procedure difficult to reproduce [12,13], i.e., the product we obtained proved to be a mixture of stereoisomers. The development in partial and total syntheses of thiasteroids has been reviewed, but steroid analogs containing sulfur in the side chain were not included in this article [14].

We have recently described a simple synthesis of carbaanalogs of steroidal sapogenins along with their biological activity [15]. Unfortunately, the carbaanalog (**5**) was obtained as an inseparable mixture of *cis/trans* isomers (note that C22 and C25 are not stereogenic centers in this compound).

## 2. Results and Discussion

### 2.1. Chemistry

The presence of a soft sulfur atom in 26-thiodiosgenin (**3**) alters its chemical, physical, and biological properties, making it, among others, very susceptible to oxidation. In fact, diosgenin (**2**) can also be oxidized with different reagents; its oxidation sites are the C5-C6 double bond, the 3β-hydroxyl group, and the carbon atoms C20 or C23. However, the introduction of sulfur in place of oxygen changes the reactivity of compound and directs the oxidation of **3** to the sulfur atom. It is well known that the oxidation of sulfides is a two-step process. In the first step, only one oxygen atom is transferred from the oxidizing agent to sulfur, forming sulfoxide. Since a new stereogenic center is generated at sulfur during this step (provided that the starting sulfide is not symmetrical), two diastereomeric sulfoxides can be formed. Further oxidation of both sulfoxides leads to a single sulfone. The oxidation of sulfoxides to sulfones is relatively easy, and sometimes it is difficult to stop the oxidation of sulfides at an intermediate stage. The simplest method of sulfoxide and sulfone synthesis is the sulfide oxidation with halogen derivatives and metal-mediated oxidative systems [16]. However, due to growing concerns about chemical pollution and environmental protection, there is a tendency to use hydrogen peroxide or peroxy acids as atom-efficient and environmentally benign oxidizing agents [16,17]. The oxidation of organic sulfides with peroxycarboxylic acids or with hydrogen peroxide, which is usually activated by conventional acidic catalysts, often leads to various side reactions such as over-oxidation (sulfoxides to sulfones), epoxidation (if a double bond is present) and Baeyer–Villiger oxidation (if an oxo group is present). After testing several methods of the chemoselective sulfide oxidation, we turned our attention to the method employing hydrogen peroxide and *N*-hydroxysuccinimide (NHS) [18]. According to the literature, sulfoxides can be obtained with this reagent in acetone under reflux conditions, without over-oxidation to sulfones, and the method is compatible with the presence of sensitive groups, including alkenes and hydroxyl groups. Nonetheless, a complex mixture of products was formed when 26-thiodiosgenin (**3**) was subjected to the described reaction conditions. In turn, when compound **3** was treated with this oxidizing system under milder conditions (50 °C, 10 h), the corresponding sulfone **8** was obtained as the only product in 70% yield (Figure 1). The formation of intermediate sulfoxides **6** and **7** was not observed, even after lowering the reaction temperature to 30 °C and reducing the reaction time. The yield of sulfone **8** then dropped to 68%.

Another reagent recommended for oxidation of sulfides to sulfoxides is m-chloroperoxybenzoic acid (MCPBA) [19]. The reaction was carried out with 1.1 equiv. of this reagent at −78 °C (dry ice/acetone bath). The progress of the reaction was monitored by TLC, which showed a single spot of a very polar product (more polar than sulfone **8**). After two hours, the reaction came to completion and was quenched with dimethylsulfide.

Although the product appeared to be single by TLC, a detailed analysis of its spectra (Figure 2) revealed that it was a mixture of sulfoxides, one of which was by far predominant. In the ^1^H NMR spectrum, a tiny signal at about 3 ppm derived from protons of the C26 methylene group of the minor sulfoxide (the C26 methylene protons of the major one appeared as a doublet at δ 2.78 and triplet at δ 2.57 ppm). Furthermore, the LC-MS analysis unambiguously confirmed the presence of trace amounts of the second sulfoxide. Unfortunately, the separation of these isomers was practically impossible, even by crystallization of crude product.

The reaction of 26-thiodiosgenin (**3**) with 2.2 equiv. of MCPBA at −40 °C afforded sulfone **8** as the major product, in addition to 10% yield of sulfoxides **6**/**7** and 2% yield of the over-oxidized product 5,6-epoxysulfone (Figure 2).

The low temperature (−78 °C) MCPBA oxidation of 26-thiodiosgenin 4-nitrobenzoate (**3a**) was also carried out with the expectation of obtaining the well-crystallizing and easily separable products. Indeed, it turned out that the separation of isomeric sulfoxides was possible not only by HPLC but also by a silica gel (230–400 mesh) gravity flow column chromatography. The less polar minor product was obtained in 9% yield, while the major one in 71% yield (Figure 3).

A detailed analysis of ^1^H NMR spectra of sulfoxides (Figure 3) can unequivocally ascribe the configuration at the sulfur atom. The diagnostic for the configuration assignment proved to be the H-16α signal. Cone shielding anisotropy generated by the sulfinyl group is similar to that of the carbonyl group. The inspection of Dreiding models, as well as molecular modeling employing the MM+ calculations (Figure 4), show that the α proton at C16 is in close proximity to the sulfinyl group (deshielding zone) in the equatorial sulfoxide **7** (configuration *R* at sulfur), while the effect of the axial *S*-oxide **6** (configuration *S* at sulfur) on this proton is negligible. As can be seen from Table 1, the H-16α signal appears at δ 4.64 in 26-thiodiosgenin (**3**), at δ 4.55–4.58 for axial sulfoxides **6** (or **6a**), and it is strongly deshielded (to δ 5.37–5.38) for equatorial sulfoxides **7** (or **7a**).

The sulfoxides are usually configurationally stable. However, in our case, the sulfoxides are stable in solid state only, but in solution (dichloromethane) a slow equilibration between the isomeric sulfoxides **6** and **7** occurs. The solution of **6a** was allowed to stand under argon at room temperature and the progress of isomerization was monitored by TLC. The equilibrium was reached within 14 days, and then the ratio of **6a**:**7a** was 3:2. The same isomeric ratio was obtained when the equatorial sulfoxide **7a** was subjected to equilibration. This result is consistent with molecular mechanics calculations which showed a slightly lower steric energy for the axial sulfoxide (Figure 4). The equilibration between sulfoxides can be reached much faster (10 min) if a catalytic amount of *p*-TsOH is added. However, when the reaction mixture was allowed to stand for a longer time, further isomerization processes occurred (presumably at C20 and C22); as a result, up to 8 isomers can be formed. The tentative mechanism of the sulfoxide equilibration is shown in Figure 4.

This unusual behavior of sulfoxides **6** and **7** is caused by an easy cleavage of the C22-S bond under acidic conditions. The cleavage leads to the sulfenic acid and the relatively stable oxocarbenium ion. The reverse reaction recovers the sulfoxide, but the configuration at the sulfur atom may be inversed. It should be noted that if the sulfoxide is treated with acid for a longer time, the C20 proton may be abstracted from the oxocarbenium ion to give the C20–C22 double bond, and then an isomerization at these carbon atoms may also occur.

On the other hand, the C22-S bond in sulfoxides **6**/**7** and sulfone **8** should be resistant to basic conditions. If so, an α-alkylation should be possible. Consequently, the starting 26-thiodiosgenin (**3**) was converted to *tert*-butyldimethylsilyl ether **3b** with TBDMS-Cl and imidazole. Then **3b** was oxidized with 1.1 equiv. of MCPBA at −78 °C affording sulfoxide **6b**. After increasing the reaction temperature to −40 °C and raising the amount of oxidant to 2.2 equiv., the corresponding sulfone **8b** was also prepared. The obtained sulfoxide **6b** and sulfone **8b** were deprotonated with n-butyllithium and then treated with 2.5 equiv. of methyl iodide (Figure 5). The reaction of sulfoxide **6b** led to the formation of the dimethylated product **10b** in 91% yield. The α-methylation of sulfone **8b** also proceeded smoothly, but only the mono substituted product **11b** was obtained. The difference in the reaction course is probably due to a larger steric hindrance at C26 in the sulfone than in the corresponding sulfoxide. The α-methylation of sulfone **8b** proved to be highly stereoselective, providing only one product **11b** in 82% yield. The S configuration at the newly formed stereogenic center at C26 was concluded on the basis of the ^1^H NMR signal of H-26, which appeared at δ 3.14 as a doublet of quartets with coupling constants *J* = 6.9 and 11.1 Hz. The latter comes from the coupling of two axial protons at C26 (alpha) and C25 (beta). It implies that the new methyl group at C26 assumed the equatorial position. The reaction of sulfone **8b** with ethyl iodide also provided the α-substituted product, albeit in slightly lower yield (76%). Attempts of α-alkylation of 26-thiodiosgenin 3-TBDMS-ether (**3b**) failed, probably due to a low acidity of the α-proton in this compound. This result could be expected given the lower acidity of dimethylsulfide (pK_a_ = 45.0) compared with dimethylsulfoxide (pK_a_ = 35.1) and dimethylsulfone (pK_a_ = 31.1).

### 2.2. Biology

#### 2.2.1. Antimicrobial Activity of Tested Compounds

Diosgenin (**1**) belongs to the group of steroidal compounds known as sapogenins, which are obtained from their glycoside forms, i.e., saponins [20]. It demonstrates various biological activities, i.a. anticancer, antioxidant, and anti-inflammatory activity [21]. In order to verify if newly synthesized sulfur analogs of diosgenin (**1**) possess antimicrobial activity, the MIC (minimal inhibition concentration) and MBC (minimal bactericidal concentration) values were estimated, using two bacterial strains: Gram-positive *Staphylococcus aureus* 8325-4 and Gram-negative *Escherichia coli* 35218. The obtained MIC and MBC values are shown in Table 2.

Diosgenin (**1**) has a strong antimicrobial activity against *S. aureus* (Gram-positive bacteria), and its growth is inhibited at the concentration of 3.9 µg/mL, whereas four-times higher concentration (15.615 µg/mL) was estimated as MBC for these bacteria. Much weaker activity was detected for diosgenin (**1**) against Gram-negative *E. coli* where MIC and MBC were 250 µg/mL and 500 µg/mL, respectively. This effect may be due to the characteristic structure of the outer membrane of Gram-negative bacteria, which contains unique component—lipopolysaccharide [22]. Exchanging the oxygen atom with a sulfur atom in 26-thiodiosgenin (**3**) performed via chemical modification of the diosgenin F-ring results in an increase of antibacterial activity for both the *S. aureus* and *E. coli*. The obtained MIC and MBC values were: 1.95 µg/mL and 3.9 µg/mL for *S. aureus*, and 62.5 µg/mL and 125 µg/mL for *E. coli*. This may be a consequence of the presence of a sulfur atom in the structure of these compounds. It is well know that sulfur functional groups are found in many pharmaceuticals, including penicillin, prevacid (lansoprazole), seroquel (quetiapine), dapsone, or sulfamethoxazole [23]. From the chemical point of view, they belong to different classes of sulfur compounds, e.g., cyclic sulfides, sulfoxides, sulfones, or sulfonamides. The oxidation of 26-thiodiosgenin (**3**) to sulfoxide or sulfone as well as their α-methylation decreased antibacterial activity of compounds **6**, **8**, **11** compared with **3**. However, their antimicrobial activity was still stronger than that of diosgenin (**1**) (see Table 2). The weakest activity has been detected for carbaanalog **5** with the MIC values 7.8 µg/mL and 500 µg/mL for *S. aureus* and *E. coli*, respectively. This is in line with our previous findings for not methylated 5- and 6-membered F-ring carbaanalogs recently described [15]. The latter compound showed antimicrobial activity against *S. aureus* 8325-4 with the MIC = 4 µg/mL and against *E. coli* with the MIC = 512 µg/mL. The carbaanalog **5** demonstrated slightly lower antimicrobial potential against *S. aureus* 8325-4 (MIC = 7.8 µg/mL), probably due to the presence of an additional 27-methyl group. In the case of *E. coli* the activity was similar (MIC = 500 µg/mL for **5** and MIC = 512 µg/mL for its non-methylated analog). The determined MIC and MBC values allowed the conclusion that 26-thiodiosgenin (**3**) demonstrated the strongest antibacterial activity against *S. aureus* 8325-4 and *E. coli* ATCC 35218 among all tested compounds. Both its *S*-oxidation and α-methylation of the oxidized derivatives decreased their antimicrobial potential.

#### 2.2.2. The Interaction of Studied Compounds with Proteins of Bacterial Cell Membranes

The results described above clearly showed that studied compounds exhibit antibacterial activity against *S. aureus* and *E. coli*. This feature may be due to the interaction between the molecule and specific component of the cell membrane, i.e., a protein or a phospholipid. In order to check if prepared thia-steroids possess the ability to interact with bacterial membranes proteins, we have analyzed fluorescence changes of tryptophan residues of membrane proteins as the marker of such interactions.

Based on fluorescence analysis, the Stern-Volmer plots were drawn (Figure 5) using equation given below (Equation (1)) [24].
(1)$$\frac{F_{0}}{F}=1+K_{SV}\left[Q\right]$$
where: *F*_0_ and *F* are the Trp fluorescence without and in the presence of quencher (studied compounds), *K_SV_*—Stern-Volmer constant, [*Q*]—concentration of the quencher.

Referring to Figure 5 it can be concluded that all compounds have an affinity for Trp^214^ residues in a hydrophobic pocket of Gram-positive (*S. aureus*) and Gram-negative (*E. coli*) bacterial membrane proteins, but the observed effect was visibly stronger for *S. aureus*. Based on the Equation (1), the Stern-Volmer constants have been calculated and are presented below (Table 3).

According to calculated *K_SV_* constants, it can be inferred that diosgenin (**1**), as well as its analogs, possess higher affinity to Trp^214^ residues localized in *S. aureus* membrane proteins in comparison with *E. coli*. These results are in good agreement with obtained MIC and MBC values. Thiodiosgenin (**3**) (*K_SV_* = (8.322 ± 0.142) × 10^4^ M^−1^ for *S. aureus* and *K_SV_* = (5.091 ± 0.152) × 10^4^ M^−1^ for *E. coli*) demonstrated higher ability to interact with membrane proteins of both used bacteria strains in comparison with diosgenin (**1**) (*K_SV_* = (5.441 ± 0.131) × 10^4^ M^−1^ and *K_SV_* = (3.384 ± 0.165) × 10^4^ M^−1^ for *S. aureus* and *E. coli*, respectively). The oxidation of thiodiosgenin (**3**) and further α-methylation of the corresponding sulfone decreased the affinity of these compounds to tryptophan residues localized in the membrane proteins of *S. aureus* and *E. coli* to the values lower than that of diosgenin (**1**) and thiodiosgenin (**3**). Despite different affinity of studied compounds to Trp^214^ residue of the bacterial membrane proteins, it can be concluded that their antibacterial activity is closely related to it. The lowest *K_SV_* in the interaction with *S. aureus* was obtained for carbaanalog **5** (*K_SV_* = (1.258 ± 0.189) × 10^4^ M^−1^). It indicated the weakest affinity of this compound to the hydrophobic pockets of tryptophan in staphylococcal membrane proteins.

In order to check whether studied compounds form complexes with bacterial cells proteins, the quenching constants (*k_q_*) have been calculated using equation below (Equation (2)) and are presented in Table 3.
(2)$$k_{q}=\frac{K_{SV}}{\tau_{0}}$$
where: *k_q_*—quenching constant, *K_SV_*—Stern-Volmer constant, *τ*_0_—fluorescence lifetime of fluorophore molecules.

The calculated *k_q_* values for all tested compounds were greater than the one for the maximum scatter collision (2 × 10^10^ M^−1^s^−1^), thus it can be deduced that investigated molecules formed complexes with *S. aureus* and *E. coli* membrane proteins.

#### 2.2.3. Cytotoxicity Study

The obtained sulfur analogs of diosgenin (**3**, **6**–**12**) were briefly tested for cytotoxicity against three cancer lines (MCF7, K562, and HeLa) and normal human retina cells RPE-1. All tested compounds have not been shown to be toxic to normal RPE-1 cells, as well as towards MCF7 and K562 (Table 4). However, sulfoxides **7** and **10**, as well as alkylated sulfones **11** and **12**, exhibited a moderate toxicity against human cervical carcinoma cells (HeLa) with IC_50_ 34.3, 32.2, 92.3 and 44.1 μM, respectively. Interestingly, the carbaanalog **5** proved also slightly cytotoxic against HeLa cancer cell line (IC_50_ 38.2 μM).

## 3. Materials and Methods

### 3.1. Chemistry

#### 3.1.1. General

The reagents were purchased from Merck, Alfa Aesar, or Acros. All solvents were freshly distilled prior to use. The dry solvents were prepared by distillation over the following drying agents: DMF (4 Å molecular sieves), THF (Na/benzophenone), CH_2_Cl_2_ (CaH_2_).

The reactions were monitored by TLC on silica gel plates 60 F254 (Merck, Darmstadt, Germany) and spots were visualized either by UV Hand Lamp (Type: NU-4; 254 nm/365 nm, 2x4W, Herolab GmbH Laborgeräte, Wiesloch, Germany) or by charring with molybdophosphoric acid/cerium(IV) sulfate in H_2_SO_4_. The reaction products were isolated by chromatographic methods using JT Baker silica gel (J.T. Baker, Phillipsburg, NJ, USA), pore size 40 Å (70–230 mesh) (column chromatography) or 60 Å (230–400 mesh) (gravity flow column chromatography and dry flash chromatography).

^1^H and ^13^C NMR (400 and 100 MHz, respectively) spectra of all compounds were recorded using Bruker Avance II spectrometer (Bruker, Fällanden, Switzerland) in CDCl_3_ and referenced to TMS (0.0 ppm) and CDCl_3_ (77.0 ppm), respectively. Only selected signals in the ^1^H NMR spectra are reported. The original ^1^H and ^13^C NMR spectra are contained in the Appendix A. Infrared spectra were recorded using Attenuated Total Reflectance (ATR) as solid samples with Nicolet 6700 FT-IR spectrometer (Thermo Fisher Scientific, Waltham, MA, USA). Mass spectra were obtained at Accurate-Mass Q-TOF LC/MS 6530 spectrometer (Agilent, Santa Clara, NJ, USA) with electrospray ionization (ESI). Melting points were determined on a Kofler bench (Boetius type, Nagema, VEB Wägetechnik Rapido Radebeul, Dresden, Germany) melting point apparatus.

26-Thiodiosgenin (**3**) [10] and carbaanalog (**5**) [15] were prepared according to literature procedures.

#### 3.1.2. Synthesis of 26-Thiodiosgenin 3β-4-Nitrobenzoate (3a)

26-Thiodiosgenin (**3**) (200 mg, 0.47 mmol) was dissolved in a mixture of dry pyridine (25 mL) and dichloromethane (10 mL), and then 4-nitrobenzoyl chloride (106 mg, 0.56 mmol) was added. The reaction mixture was stirred at room temperature. After completion of the reaction (7 days), it was poured into water, and extracted with dichloromethane (3 × 100 mL). The combined organic extracts were dried over anhydrous Na_2_SO_4_, and the solvent was evaporated in vacuo. The crude product was purified by dry flash chromatography with a hexane/ethyl acetate (99:1) mixture to afford ester **3a** (245 mg, 91%).

Compound **3a**: colorless crystals (hexane/ethyl acetate); mp 257–258 °C; IR (ATR) ν_max_ 1601, 1520, 1449, 1337, 1270, 1106, 1013 cm^−1^; ^1^H NMR (CDCl_3_, 400 MHz) *δ* 8.28 (2H, d, *J* = 8.7 Hz, H-Ar), 8.21 (2H, d, *J* = 8.7 Hz, H-Ar), 5.44 (1H, m, H-6), 4.90 (1H, m, H-3α), 4.65 (1H, m, H-16), 2.55 (1H, t, *J* = 12.6 Hz, H-26α), 2.30 (1H, d, *J* = 12.6 Hz, H-26β), 1.10 (3H, s, H-19), 1.03 (3H, d, *J* = 6.9 Hz, H-21), 0.94 (3H, d, *J* = 6.5 Hz, H-27), 0.83 (3H, s, H-18); ^13^C NMR (CDCl_3_, 100 MHz) δ 164.0 (C), 150.4 (C), 139.3 (C), 136.2 (C), 130.6 (2 × CH), 123.4 (2 × CH), 122.9 (CH), 97.5 (C), 81.6 (CH), 75.7 (CH), 62.8 (CH), 56.5 (CH), 49.9 (CH), 44.4 (CH), 40.3 (C), 39.7 (CH_2_), 38.5 (CH_2_), 38.0 (CH_2_), 36.9 (C), 36.7 (CH_2_), 33.3 (CH), 32.1 (CH_2_), 32.0 (CH_2_), 31.7 (CH_2_), 31.4 (CH), 31.4 (CH_2_), 27.7 (CH_2_), 22.5 (CH_3_), 20.8 (CH_2_), 19.4 (CH_3_), 16.5 (CH_3_), 16.2 (CH_3_); HRMS *m*/*z* 580.3104 (calcd for C_34_H_46_NO_5_S^+^, 580.3091).

#### 3.1.3. Preparation of 26-Thiodiosgenin 3β-t-Butyldimethylsilyl Ether (3b)

Imidazole (48 mg, 0.7 mmol) and *t*-butyldimethylsilyl chloride (106 mg, 0.7 mmol) were added to a solution of 26-thiodiosgenin (**3**) (200 mg, 0.47 mmol) in dry dimethylformamide (20 mL). The reaction mixture was stirred for 24 h at room temperature, then poured into water, and extracted with diethyl ether (3 × 100 mL). The combined organic extracts were dried over anhydrous Na_2_SO_4_, and concentrated under reduced pressure. The crude product was purified by dry flash chromatography with hexane/ethyl acetate (97:3) elution to afford ether **3b** (207 mg, 82%).

Compound **3b**: colorless crystals (hexane/ethyl acetate); mp 235–237 °C; IR (ATR) ν_max_ 1458, 1373, 1248, 1092, 1014 cm^−1^; ^1^H NMR (CDCl_3_, 400 MHz) *δ* 5.32 (1H, m, H-6), 4.64 (1H, dd, *J* = 15.4 Hz, *J* = 7.4 Hz, H-16), 3.49 (1H, m, H-3α), 2.55 (1H, dd, *J* = 12.7 Hz, *J* = 11.7 Hz, H-26α), 2.30 (1H, d, *J* = 12.7 Hz, H-26β), 1.027 (3H, d, *J* = 6.8 Hz, H-21), 1.026 (3H, s, H-19), 0.94 (3H, d, *J* = 6.5 Hz, H-27), 0.90 (9H, s, *t*-Bu-Si), 0.81 (3H, s, H-18), 0.07 (6H, s, (CH_3_)_2_Si); ^13^C NMR (CDCl_3_, 100 MHz) *δ* 141.6 (C), 120.9 (CH), 97.5 (C), 81.7 (CH), 72.6 (CH), 62.8 (CH), 56.7 (CH), 50.1 (CH), 44.4 (CH), 42.8 (CH_2_), 40.3 (C), 39.8 (CH_2_), 38.5 (CH_2_), 37.3 (CH_2_), 36.7 (C), 33.3 (CH), 32.10 (CH_2_), 32.09 (CH_2_), 32.06 (CH_2_), 31.7 (CH_2_), 31.5 (CH), 31.4 (CH_2_), 25.9 (3 x CH_3_), 22.5 (CH_3_), 20.8 (CH_2_), 19.4 (CH_3_), 18.3 (C), 16.5 (CH_3_), 16.2 (CH_3_), −4.6 (2 × CH_3_); HRMS *m*/*z* 545.3829 (calcd for C_33_H_57_O_2_SSi^+^, 545.3843).

#### 3.1.4. General Procedure for S-Oxides Formation

Steroidal sulfide (**3**, **3a** or **3b**) (0.23, mmol) was dissolved in dry dichloromethane (20 mL), cooled to −78 °C (dry ice/acetone cooling bath), and *m*-chloroperoxybenzoic acid (56 mg, 0.25 mmol) was added. The solution was stirred at −78 °C for 2–3 h and monitored by TLC. When the reaction was completed, dimethyl sulfide (0.1 mL, 1.3 mmol) was added. The reaction mixture was poured into saturated aqueous solution of NaHCO_3_, and extracted with dichloromethane (3 × 100 mL). The combined organic extracts were dried over anhydrous Na_2_SO_4_, and evaporated to dryness in vacuo. The residue was subjected to gravity flow column chromatography on silica gel 60 Å (230–400 mesh), which resulted in separation of compounds **6**/**7** (as a mixture), **6a** and **7a**, or **8b**.

Compound **6a**: was eluted with benzene/ethyl acetate (1:1) mixture in 71% yield. Colorless crystals (benzene/ethyl acetate); mp 150–151 °C; IR (ATR) ν_max_ 1600, 1521, 1452, 1345, 1270, 1168, 1107, 1017 cm^−1^; ^1^H NMR (CDCl_3_, 400 MHz) *δ* 8.24 (2H, d, *J* = 9.0 Hz, H-Ar), 8.17 (2H, d, *J* = 9.0 Hz, H-Ar), 5.39 (1H, m, H-6), 4.84 (1H, m, H-3α), 4.55 (1H, m, H-16), 2.74 (1H, d, *J* = 13.3 Hz, H-26β), 2.53 (1H, t, *J* = 13.3 Hz, H-26α), 1.32 (3H, d, *J* = 7.2 Hz, H-21), 1.06 (3H, s, H-19), 0.94 (3H, d, *J* = 6.7 Hz, H-27), 0.80 (3H, s, H-18); ^13^C NMR (CDCl_3_, 100 MHz) *δ* 163.9 (C), 150.3 (C), 139.3 (C), 136.0 (C), 130.5 (2 × CH), 123.3 (2 × CH), 122.5 (CH), 99.6 (C), 84.2 (CH), 75.5 (CH), 63.6 (CH), 56.2 (CH), 49.7 (CH), 49.0 (CH_2_), 43.7 (CH), 40.8 (C), 39.1 (CH_2_), 37.9 (CH_2_), 36.8 (CH_2_), 36.6 (C), 32.5 (CH_2_), 31.8 (CH_2_), 31.2 (CH), 29.7 (CH_2_), 27.65 (CH_2_), 27.60 (CH_2_), 21.3 (CH_3_), 20.6 (CH_2_), 19.9 (CH), 19.2 (CH_3_), 16.7 (CH_3_), 16.1 (CH_3_); HRMS *m*/*z* 596.3020 (calcd for C_34_H_46_NO_6_S^+^, 596.3040).

Compound **7a**: was eluted with benzene/ethyl acetate (1:1) mixture in 9% yield. Colorless crystals (benzene/ethyl acetate); mp 175–176 °C; IR (ATR) ν_max_ 1522, 1453, 1342, 1268, 1163, 1108, 1038 cm^−1^; ^1^H NMR (CDCl_3_, 400 MHz) *δ* 8.28 (2H, d, *J* = 9.0 Hz, H-Ar), 8.20 (2H, d, *J* = 9.0 Hz, H-Ar), 5.43 (1H, m, H-6), 5.38 (1H, m, H-16), 4.89 (1H, m, H-3α), 3.06-2.98 (2H, m, H-26), 1.46 (3H, d, *J* = 7.0 Hz, H-21), 1.09 (3H, s, H-19), 1.05 (3H, d, *J* = 6.6 Hz, H-27), 0.82 (3H, s, H-18); ^13^C NMR (CDCl_3_, 100 MHz) *δ* 164.0 (C), 150.4 (C), 139.2 (C), 136.2 (C), 130.6 (2 × CH), 123.4 (2 × CH), 122.8 (CH), 104.4 (C), 87.3 (CH), 75.6 (CH), 63.0 (CH), 56.0 (CH), 52.4 (CH_2_), 49.8 (CH), 43.9 (CH), 40.7 (C), 39.4 (CH_2_), 38.0 (CH_2_), 36.8 (CH_2_), 36.7 (C), 33.6 (CH_2_), 33.2 (CH_2_), 32.0 (CH_2_), 31.3 (CH), 30.0 (CH_2_), 29.7 (CH), 27.7 (CH_2_), 21.5 (CH_3_), 20.8 (CH_2_), 19.3 (CH_3_), 16.4 (2 x CH_3_); HRMS *m*/*z* 596.3023 (calcd for C_34_H_46_NO_6_S^+^, 596.3040).

Compound **6b**: was eluted with hexane/ethyl acetate (7:3) mixture in 69% yield. Colorless crystals (hexane/ethyl acetate); mp 205–207 °C; IR (ATR) ν_max_ 1571, 1455, 1376, 1248, 1083, 1031 cm^−1^; ^1^H NMR (CDCl_3_, 400 MHz) *δ* 5.32 (1H, m, H-6), 4.58 (1H, m, H-16), 3.49 (1H, m, H-3α), 2.78 (1H, d, *J* = 13.0 Hz, H-26β), 2.57 (1H, t, *J* = 13.0 Hz, H-26α), 1.36 (3H, d, *J* = 7.3 Hz, H-21), 1.03 (3H, s, H-19), 0.98 (3H, d, *J* = 6.7 Hz, H-27), 0.90 (9H, s, *t*-Bu-Si), 0.82 (3H, s,H-18), 0.07 (6H, s, (CH_3_)_2_Si); ^13^C NMR (CDCl_3_, 100 MHz) *δ* 141.7 (C), 120.6 (CH), 99.7 (C), 84.4 (CH), 72.5 (CH), 63.8 (CH), 56.5 (CH), 50.0 (CH), 49.1 (CH_2_), 43.8 (CH), 42.7 (CH_2_), 40.9 (C), 39.3 (CH_2_), 37.3 (CH_2_), 36.7 (C), 32.6 (CH_2_), 31.99 (CH_2_), 31.97 (CH_2_), 31.4 (CH), 29.8 (CH_2_), 27.8 (CH_2_), 25.9 (3 × CH_3_), 21.4 (CH_3_), 20.7 (CH_2_), 20.0 (CH), 19.4 (CH_3_), 18.2 (C), 16.8 (CH_3_), 16.2 (CH_3_), −4.6 (2 × CH_3_); HRMS *m*/*z* 561.3784 (calcd for C_33_H_57_O_3_SSi^+^, 561.3792).

The mixture of compounds **6** and **7** proved inseparable. These compounds were obtained in their pure forms by removing *p*-nitrobenzoyl groups from separated compounds **6a** and **7a**, what is described below.

#### 3.1.5. General Procedure for Removing the 3β-4-Nitrobenzoyl Group

To a solution of steroidal sulfoxide (**6a** or **7a**) (50 mg) (0.08 mmol) in dry methanol (10 mL) NaOH (80 mg, 2 mmol) was added. The reaction was stirred for 24 h. Then, the solvent was evaporated in vacuo, the crude product was dissolved in dichloromethane and washed by water. The organic extract was dried over anhydrous Na_2_SO_4_, and concentrated under reduced pressure. The crude product (**6** or **7**) was purified by column chromatography on silica gel with hexane/ethyl acetate (3:7) mixture elution.

Compound **6**: 89% yield; colorless crystals (hexane/ethyl acetate); mp 143–146 °C; IR (ATR) ν_max_ 3331, 1449, 1377, 1348, 1164, 1139, 1052, 1017 cm^−1^; ^1^H NMR (CDCl_3_, 400 MHz) *δ* 5.35 (1H, m, H-6), 4.58 (1H, m, H-16), 3.53 (1H, m, H-3α), 2.78 (1H, d, *J* = 13.0 Hz, H-26β), 2.57 (1H, t, *J* = 13.0 Hz, H-26α), 1.36 (3H, d, *J* = 7.3 Hz, H-21), 1.04 (3H, s, H-19), 0.98 (3H, d, *J* = 6.7 Hz, H-27), 0.83 (3H, s, H-18); ^13^C NMR (CDCl_3_, 100 MHz) *δ* 140.9 (C), 121.1 (CH), 99.7 (C), 84.4 (CH), 71.6 (CH), 63.8 (CH), 56.5 (CH), 50.0 (CH), 49.2 (CH_2_), 43.9 (CH), 42.2 (CH_2_), 40.9 (C), 39.3 (CH_2_), 37.2 (CH_2_), 36.6 (C), 32.6 (CH_2_), 31.9 (CH_2_), 31.6 (CH_2_), 31.4 (CH), 29.8 (CH_2_), 27.8 (CH_2_), 21.4 (CH_3_), 20.7 (CH_2_), 20.0 (CH), 19.4 (CH_3_), 16.8 (CH_3_), 16.3 (CH_3_); HRMS *m*/*z* 447.2921 (calcd for C_27_H_43_O_3_S^+^, 447.2927).

Compound **7**: 87% yield; colorless crystals (hexane/ethyl acetate); mp 165–166 °C; IR (ATR) ν_max_ 3420, 1455, 1377, 1341, 1156, 1138, 1074, 1033, 1011 cm^−1^; ^1^H NMR (CDCl_3_, 400 MHz) *δ* 5.40–5.34 (2H, m, H-6, H-16), 3.53 (1H, m, H-3α), 3.07-2.98 (2H, m, H-26), 1.46 (3H, d, *J* = 8.8 Hz, H-21), 1.05 (3H, d, *J* = 6.6 Hz, H-27), 1.03 (3H, s, H-19), 0.82 (3H, s, H-18); ^13^C NMR (CDCl_3_, 100 MHz) *δ* 140.9 (C), 121.3 (CH), 104.5 (C), 87.4 (CH), 71.7 (CH), 63.1 (CH), 56.2 (CH), 52.6 (CH_2_), 50.1 (CH), 44.1 (CH), 42.3 (CH_2_), 40.7 (C), 39.6 (CH_2_), 37.3 (CH_2_), 36.7 (C), 33.7 (CH_2_), 33.3 (CH_2_), 32.0 (CH_2_), 31.7 (CH_2_), 31.5 (CH), 30.1 (CH_2_), 29.7 (CH), 21.5 (CH_3_), 20.9 (CH_2_), 19.4 (CH_3_), 16.4 (CH_3_), 16.3 (CH_3_); HRMS *m*/*z* 447.2984 (calcd for C_27_H_43_O_3_S^+^, 447.2927).

#### 3.1.6. Procedures for Preparation of Sulfones

##### Procedure 1 (with H_2_O_2_ and NHS)

30% hydrogen peroxide (45 µL, 0.4 mmol) and NHS (*N*-hydroxysuccinimide) (23 mg, 0.2 mmol) were added to a solution of 26-thiodiosgenin (**3**) (40 mg, 0.1 mmol) in acetone (5 mL). The mixture was heated at 50 °C for 10 h. Then, the reaction mixture was poured into aqueous solution of NaHSO_3_, and extracted with ethyl acetate (3 × 50 mL). The combined organic extracts were dried over anhydrous Na_2_SO_4_, and evaporated to dryness in vacuo. The crude product was purified by column chromatography on silica gel with hexane/ethyl acetate (3:1) elution to afford sulfone (**8**) (230 mg, 70%).

Compound **8**: was eluted with hexane/ethyl acetate (3:1) mixture in 70% yield. Colorless crystals (hexane/ethyl acetate); mp 205–207 °C; IR (ATR) ν_max_ 3573, 3431, 1464, 1291, 1261, 1050, 1021 cm^−1^; ^1^H NMR (CDCl_3_, 400 MHz) *δ* 5.36 (1H, m, H-6), 5.10 (1H, m, H-16), 3.54 (1H, m, H-3α), 3.16 (1H, t, *J* = 13.2 Hz, H-26α), 2.55 (1H, d, *J* = 13.2 Hz, H-26β), 1.42 (3H, d, *J* = 7.3 Hz, H-21), 1.04 (3H, s, H-19), 1.02 (3H, d, *J* = 6.7 Hz, H-27), 0.80 (3H, s, H-18); ^13^C NMR (CDCl_3_, 100 MHz) *δ* 140.8 (C), 121.2 (CH), 103.3 (C), 85.5 (CH), 71.6 (CH), 63.9 (CH), 56.3 (CH), 55.2 (CH_2_), 49.9 (CH), 42.9 (CH), 42.2 (CH_2_), 40.8 (C), 39.3 (CH_2_), 37.2 (CH_2_), 36.6 (C), 35.2 (CH_2_), 32.6 (CH_2_), 31.9 (CH_2_), 31.6 (CH_2_), 31.4 (CH), 30.9 (CH), 29.1 (CH_2_), 21.1 (CH_3_), 20.8 (CH_2_), 19.4 (CH_3_), 16.2 (CH_3_), 16.0 (CH_3_); HRMS *m*/*z* 463.2856 (calcd for C_27_H_43_O_4_S^+^, 463.2877).

##### Procedure 2 (with MCPBA)

Steroidal sulfide (**3** or **3b**) (0.23 mmol) was dissolved in dry dichloromethane (20 mL) cooled to −40 °C (dry ice/acetonitrile cooling bath) and m-chloroperoxybenzoic acid (112 mg, 0.50 mmol) was added. The solution was stirred at −40 °C for 2–3 h and monitored by TLC. When the reaction was completed, dimethyl sulfide (0.2 mL, 2.6 mmol) was added. Then, the reaction mixture was poured into saturated aqueous solution of NaHCO_3_, and extracted with dichloromethane (3 × 100 mL). The combined organic extracts were dried over anhydrous Na_2_SO_4_, and evaporated to dryness in vacuo. The residue was chromatographed on a silica gel column to afford pure compounds **8** or **8b** from **3** and **3b**, respectively.

In the reaction of **3**, in addition to **8** (65%), small amounts of 5,6-epoxides **9** (2%) and sulfoxides **6**/**7** (10%) were isolated. Compounds **6**/**7** and **8** were described above. Epoxides **9** were eluted with hexane/ethyl acetate (6:4) as a mixture of 5,6α and 5,6β epimers in the ratio of 7:3. Only major product (5,6α-epoxide) is described below.

5,6α-Epoxide **9**: IR (ATR) ν_max_ 3698, 3626, 3589, 3523, 1572, 1450, 1290, 1126, 1055 cm^−1^; ^1^H NMR (CDCl_3_, 400 MHz) *δ* 5.08 (1H, m, H-16), 3.93 (1H, m, H-3α), 3.15 (1H, t, *J* = 12.7 Hz, H-26α), 2.92 (1H, d, *J* = 4.4 Hz, H-6), 2.64 (1H, d, *J* = 12.7 Hz, H-26β), 1.40 (3H, d, *J* = 7.3 Hz, H-21), 1.09 (3H, s, H-19), 1.01 (3H, d, *J* = 6.8 Hz, H-27), 0.74 (3H, s, H-18); ^13^C NMR (CDCl_3_, 100 MHz) *δ* 103.3 (C), 85.3 (CH), 68.7 (CH), 65.6 (C), 63.7 (CH), 58.9 (CH), 56.5 (CH), 55.2 (CH_2_), 42.9 (CH), 42.3 (CH), 40.8 (C), 39.7 (CH_2_), 38.9 (CH_2_), 35.2 (CH_2_), 35.0 (C), 32.4 (CH_2_), 32.3 (CH_2_), 31.0 (CH_2_), 30.9 (CH), 29.5 (CH), 29.2 (CH_2_), 28.9 (CH_2_), 21.1 (CH_3_), 20.4 (CH_2_), 16.2 (CH_3_), 16.03 (CH_3_), 15.95 (CH_3_); HRMS *m*/*z* 479.2822 (calcd for C_27_H_43_O_5_S^+^, 479.2826).

Compound **8b**: was eluted with hexane/ethyl acetate (3:1) in 63% yield. Colorless crystals (hexane/ethyl acetate); mp 228–230 °C; IR (ATR) ν_max_ 1454, 1374, 1284, 1249, 1081 cm^−1^; ^1^H NMR (CDCl_3_, 400 MHz) *δ* 5.31 (1H, m, H-6), 5.10 (1H, m, H-16), 3.48 (1H, m, H-3α), 3.16 (1H, t, *J* = 13.2 Hz, H-26a), 2.63 (1H, dd, *J* = 13.2 Hz, *J* = 1.6 Hz, H-26b), 1.41 (3H, d, *J* = 7.3 Hz, H-21), 1.02 (3H, s, H-19), 1.01 (3H, d, *J* = 6.6 Hz, H-27), 0.89 (9H, s, t-Bu-Si), 0.79 (3H, s, H-18), 0.06 (6H, s, (CH_3_)_2_Si); ^13^C NMR (CDCl_3_, 100 MHz) *δ* 141.6 (C), 120.7 (CH), 103.3 (C), 85.5 (CH), 72.4 (CH), 63.8 (CH), 56.4 (CH), 55.2 (CH_2_), 49.9 (CH), 42.9 (CH), 42.7 (CH_2_), 40.7 (C), 39.3 (CH_2_), 37.3 (CH_2_), 36.6 (C), 35.1 (CH_2_), 32.6 (CH_2_), 32.0 (CH_2_), 31.9 (CH_2_), 31.4 (CH), 30.9 (CH), 29.1 (CH_2_), 25.9 (3 × CH_3_), 21.1 (CH_3_), 20.7 (CH_2_), 19.4 (CH_3_), 18.2 (C), 16.2 (CH_3_), 16.0 (CH_3_), −4.6 (2 × CH_3_); HRMS *m*/*z* 577.3755 (calcd for C_33_H_57_O_4_SSi^+^, 577.3741).

#### 3.1.7. General Procedure for α-Alkylation

n-BuLi (0.08 mL, 2.5 M in hexane, 0.2 mmol) was added to a solution of steroidal sulfoxide or sulfone (**6b** or **8b**) (0.08 mmol) in dry THF (10 mL), and the mixture was stirred at room temperature for 15 min. After this time, alkyl iodide (CH_3_I or C_2_H_5_I) (0.2 mmol) was added and stirring was continued for 45–90 min. Then, the reaction mixture was poured into water, and extracted with dichloromethane (3 × 50 mL). The combined organic extracts were dried over anhydrous Na_2_SO_4_, and evaporated to dryness in vacuo. The residue was subjected to chromatography on a silica gel column to afford pure compounds **10b**, **11b,** or **12b**.

Compound **10b**: was eluted with hexane/ethyl acetate (3:1) mixture in 91% yield. Colorless crystals (hexane/ethyl acetate); mp 198–200 °C; IR (ATR) ν_max_ 1426, 1359, 1219, 1091 cm^−1^; ^1^H NMR (CDCl_3_, 400 MHz) *δ* 5.31 (1H, m, H-6), 4.65 (1H, m, H-16), 3.48 (1H, m, H-3α), 1.32 (3H, d, *J* = 7.4 Hz, H-21), 1.31 (3H, s, CH_3_-C26), 1.23 (3H, s, CH_3_-C26), 1.02 (3H, s, H-19), 0.90 (3H, d, *J* = 6.9 Hz, H-27), 0.89 (9H, s, t-Bu-Si), 0.81 (3H, s, H-18), 0.06 (6H, s, (CH_3_)_2_Si); ^13^C NMR (CDCl_3_, 100 MHz) *δ* 141.7 (C), 120.6 (CH), 103.2 (C), 84.0 (CH), 72.5 (CH), 63.5 (CH), 57.5 (C), 56.5 (CH), 50.1 (CH), 46.0 (CH), 42.8 (CH_2_), 40.8 (C), 39.5 (CH_2_), 37.3 (CH_2_), 36.7 (C), 32.4 (CH_2_), 32.0 (2 × CH_2_), 31.3 (CH), 30.7 (CH), 28.5 (CH_2_), 27.4 (CH_2_), 26.2 (CH_3_), 25.9 (3 × CH_3_), 20.8 (CH_2_), 19.4 (CH_3_), 18.2 (C), 16.7 (CH_3_), 16.2 (CH_3_), 15.9 (CH_3_), 15.6 (CH_3_), −4.6 (2 x CH_3_); HRMS *m*/*z* 589.4089 (calcd for C_35_H_61_O_3_SSi^+^, 589.4105).

Compound **11b**: was eluted with hexane/ethyl acetate (93:7) mixture in 82% yield. Colorless crystals (hexane/ethyl acetate); mp 198–200 °C; IR (ATR) ν_max_ 1459, 1375, 1290, 1251, 1123, 1081 cm^−1^; ^1^H NMR (CDCl_3_, 400 MHz) *δ* 5.31 (1H, m, H-6), 5.07 (1H, m, H-16), 3.49 (1H, m, H-3α), 3.14 (1H, dk, *J* = 11.1 Hz, *J* = 6.9 Hz, H-26α), 1.42 (3H, d, *J* = 7.4 Hz, H-21), 1.29 (3H, d, *J* = 6.9 Hz, CH_3_-C26), 1.02 (3H, s, H-19), 0.98 (3H, d, *J* = 6.6 Hz, H-27), 0.89 (9H, s, t-Bu-Si), 0.80 (3H, s, H-18), 0.06 (6H, s, (CH_3_)_2_Si); ^13^C NMR (CDCl_3_, 100 MHz) *δ* 141.6 (C), 120.7 (CH), 103.5 (C), 85.5 (CH), 72.5 (CH), 63.8 (CH), 57.9 (CH), 56.4 (CH), 50.0 (CH), 43.4 (CH), 42.8 (CH_2_), 40.7 (C), 39.3 (CH_2_), 37.3 (CH_2_), 36.7 (C), 35.9 (CH), 34.8 (CH_2_), 32.6 (CH_2_), 32.03 (CH_2_), 31.98 (CH_2_), 31.5 (CH), 29.8 (CH_2_), 25.9 (3 × CH_3_), 20.7 (CH_2_), 19.4 (CH_3_), 18.9 (CH_3_), 18.2 (C), 16.2 (CH_3_), 16.0 (CH_3_), 7.3 (CH_3_), −4.6 (2 × CH_3_); HRMS *m*/*z* 591.3928 (calcd for C_34_H_59_O_4_SSi^+^, 591.3898).

Compound **12b**: was eluted with hexane/ethyl acetate (99:1) mixture in 76% yield. Colorless crystals (hexane/ethyl acetate); mp 215–217 °C; IR (ATR) ν_max_ 1457, 1375, 1286, 1254, 1082 cm^−1^; ^1^H NMR (CDCl_3_, 400 MHz) *δ* 5.31 (1H, m, H-6), 5.07 (1H, m, H-16), 3.49 (1H, m, H-3α), 2.99 (1H, dt, *J* = 11.4 Hz, *J* = 4.6 Hz, H-26α), 1.41 (3H, d, *J* = 7.3 Hz, H-21), 1.13 (3H, t, *J* = 7.6 Hz, CH_3_CH_2_-C26), 1.02 (3H, s, H-19), 1.01 (3H, d, *J* = 6.5 Hz, H-27), 0.89 (9H, s, t-Bu-Si), 0.79 (3H, s, H-18), 0.06 (6H, s, (CH_3_)_2_Si); ^13^C NMR (CDCl_3_, 100 MHz) *δ* 141.6 (C), 120.7 (CH), 103.7 (C), 85.4 (CH), 72.5 (CH), 63.8 (CH), 62.7 (CH), 56.4 (CH), 50.0 (CH), 43.4 (CH), 42.8 (CH_2_), 40.7 (C), 39.4 (CH_2_), 37.3 (CH_2_), 36.7 (C), 34.8 (CH_2_), 33.8 (CH), 32.6 (CH_2_), 32.03 (CH_2_), 31.98 (CH_2_), 31.4 (CH), 29.9 (CH_2_), 25.9 (3 × CH_3_), 20.8 (CH_2_), 19.4 (CH_3_), 19.0 (CH_3_), 18.2 (C), 16.6 (CH_2_), 16.2 (CH_3_), 16.0 (CH_3_), 11.8 (CH_3_), −4.6 (2 × CH_3_); HRMS *m*/*z* 605.4031 (calcd for C_35_H_61_O_4_SSi^+^, 605.4054).

#### 3.1.8. General Procedure for the Removal of the TBDMS Group

Steroidal 3β-*t*-butyldimethylsilyl ether (**10b**, **11b**, or **12b**) (0.08 mmol) was dissolved in dry THF (5 mL) and then tetra-n-butylammonium fluoride (0.8 mL, 1 M in THF, 0.8 mmol) was added dropwise. After stirring at room temperature for 2 h, the reaction mixture was poured into water and extracted with diethyl ether (3 × 50 mL). The combined organic layers were dried over Na_2_SO_4_, and evaporated in vacuo. The residue was purified by column chromatography on silica gel afforded compound **10**, **11**, or **12**.

Compound **10**: was eluted with hexane/ethyl acetate (7:13) mixture as a white amorphous solid in 87% yield. IR (ATR) ν_max_ 3338, 1445, 1346, 1250, 1139, 1062, 1014 cm^−1^; ^1^H NMR (CDCl_3_, 400 MHz) *δ* 5.35 (1H, m, H-6), 4.66 (1H, m, H-16), 3.53 (1H, m, H-3α), 1.33 (3H, d, *J* = 7.4 Hz, H-21), 1.32 (3H, s, CH_3_-C26), 1.23 (3H, s, CH_3_-C26), 1.04 (3H, s, H-19), 0.91 (3H, d, *J* = 6.9 Hz, H-27), 0.82 (3H, s, H-18); ^13^C NMR (CDCl_3_, 100 MHz) *δ* 141.0 (C), 121.2 (CH), 103.2 (C), 84.0 (CH), 71.7 (CH), 63.6 (CH), 57.5 (C), 56.6 (CH), 50.1 (CH), 46.1 (CH), 42.3 (CH_2_), 40.8 (C), 39.6 (CH_2_), 37.2 (CH_2_), 36.7 (C), 32.5 (CH_2_), 31.7 (2 × CH_2_), 31.4 (CH), 30.7 (CH), 28.6 (CH_2_), 27.4 (CH_2_), 26.2 (CH_3_), 20.8 (CH_2_), 19.4 (CH_3_), 16.7 (CH_3_), 16.2 (CH_3_), 15.9 (CH_3_), 15.6 (CH_3_); HRMS *m*/*z* 475.3253 (calcd for C_29_H_47_O_3_S^+^, 475.3240).

Compound **11**: was eluted with hexane/ethyl acetate (3:1) mixture in 92% yield. Colorless crystals (hexane/ethyl acetate); mp 225–227 °C; IR (ATR) ν_max_ 3514, 3355, 1452, 1378, 1348, 1290, 1279, 1117, 1049 cm^−1^; ^1^H NMR (^1^H NMR (CDCl_3_, 400 MHz) *δ* 5.35 (1H, m, H-6), 5.07 (1H, m, H-16), 3.53 (1H, m, H-3α), 3.14 (1H, dq, *J* = 11.1 Hz, *J* = 7.0 Hz, H-26α), 1.43 (3H, d, *J* = 7.4 Hz, H-21), 1.29 (3H, d, *J* = 7.0 Hz, CH_3_-C26), 1.03 (3H, s, H-19), 0.98 (3H, d, *J* = 6.6 Hz, H-27), 0.80 (3H, s, H-18); ^13^C NMR (CDCl_3_, 100 MHz) *δ* 140.9 (C), 121.2 (CH), 103.6 (C), 85.5 (CH), 71.7 (CH), 63.9 (CH), 57.9 (CH), 56.4 (CH), 49.9 (CH), 43.5 (CH), 42.3 (CH_2_), 40.8 (C), 39.4 (CH_2_), 37.2 (CH_2_), 36.6 (C), 35.9 (CH), 34.9 (CH_2_), 32.6 (CH_2_), 32.0 (CH_2_), 31.6 (CH_2_), 31.5 (CH), 29.8 (CH_2_), 20.8 (CH_2_), 19.4 (CH_3_), 18.9 (CH_3_), 16.2 (CH_3_), 16.0 (CH_3_), 7.3 (CH_3_); HRMS *m*/*z* 477.3039 (calcd for C_28_H_45_O_4_S^+^, 477.3033).

Compound **12**: was eluted with hexane/ethyl acetate (3:1) mixture in 90% yield. Colorless crystals (hexane/ethyl acetate); mp 210–212 °C; IR (ATR) ν_max_ 3503, 1450, 1377, 1349, 1276, 1225, 1152, 1117, 1072, 1051 cm^−1^; ^1^H NMR (CDCl_3_, 400 MHz) *δ* 5.35 (1H, m, H-6), 5.07 (1H, m, H-16), 3.53 (1H, m, H-3α), 2.99 (1H, dt, *J* = 11.1 Hz, *J* = 4.6 Hz, H-26α), 1.41 (3H, d, *J* = 7.2 Hz, H-21), 1.13 (3H, t, *J* = 7.5 Hz, CH_3_CH_2_-C26), 1.03 (3H, s, H-19), 1.02 (3H, d, *J* = 6.8 Hz, H-27), 0.80 (3H, s, H-18); ^13^C NMR (CDCl_3_, 100 MHz) *δ* 140.8 (C), 121.2 (CH), 103.7 (C), 85.4 (CH), 71.7 (CH), 63.7 (CH), 62.7 (CH), 56.3 (CH), 49.9 (CH), 43.4 (CH), 42.2 (CH_2_), 40.7 (C), 39.3 (CH_2_), 37.1 (CH_2_), 36.6 (C), 34.8 (CH_2_), 33.8 (CH), 32.6 (CH_2_), 31.9 (CH_2_), 31.6 (CH_2_), 31.4 (CH), 29.9 (CH_2_), 20.8 (CH_2_), 19.4 (CH_3_), 19.0 (CH_3_), 16.6 (CH_2_), 16.2 (CH_3_), 16.0 (CH_3_), 11.8 (CH_3_); HRMS *m*/*z* 491.3197 (calcd for C_29_H_47_O_4_S^+^, 491.3190).

### 3.2. Biology

#### 3.2.1. Determination of Antimicrobial Activity

The antibacterial activity of tested compounds was assessed by monitoring the cell growth of *Staphylococcus aureus* 8325-4 and *Escherichia coli* ATCC 35218 using the broth microdilution method, conducted according to the National Committee for Clinical Laboratory Standards. First, compounds were dissolved in methanol and the solution was added to Mueller Hinton broth (MHB) for bacteria to give a final concentration of 2000 µg/mL. The samples were then serially two-fold diluted in bullion to obtain concentration ranging from 1000 to 0.12 μg/mL in 96-well microtiter plate with the final volumes of 100 μL. Next, 100 μL of bacteria suspension was injected into each well. The final bacteria cell concentration was 1 × 10^6^ colony-forming units per ml (CFU/mL). The plates were incubated at 37 °C for 24 h. The MIC value was determined as the lowest concentration of an antibacterial agent that inhibited bacterial growth, as indicated by the absence of turbidity. The MBC value was determined as the lowest concentration of antibacterial agents for which no bacterial growth on the plates was observed [15].

#### 3.2.2. Fluorescence Analysis of Diosgenin and Their Derivatives Interaction with Bacterial Cell Membranes

Bacteria strains: *S. aureus* and *E. coli* grown overnight at 37 °C in Mueller Hinton (MH) broth with shaking at 200 rpm. Next, bacteria suspensions were centrifuged (2300× *g*, 15 min) and resuspended in PBS after removing the supernatant. For experiments, bacteria suspensions (OD_600_ = 0.1 in C = 10 mM PBS buffer, pH = 7.4) have been used in the presence (concentration range 0.5–2.0 µM) and without the studied compounds. The measurements were performed in quartz cuvette (1 cm × 1 cm). Fluorescence was monitored at the λ_exc._ = 295 nm and λ_em._ = 350 nm from tryptophan (Trp^214^) residues in bacterial membrane proteins. The studies were carried out using Perkin-Elmer LS-55B (Perkin-Elmer, Pontyclun, UK) spectrofluorometer [25].

#### 3.2.3. Cytotoxicity Tests

The studied compounds were evaluated for cytotoxicity in human cancer cell lines (cervical carcinoma HeLa, chronic myelogenous leukemia K562 and breast adenocarcinoma MCF7) and normal human retina cells RPE-1 (ca. 5.0 × 10^4^ cells·mL^−1^) after 72 h of treatment using resazurin (Merck/MilliporeSigma, St. Louis, MO, USA) as described earlier for Calcein AM dye [26]. The IC_50_ values (µM) obtained from at least three independent experiments in triplicates are shown in Table 3.

## 4. Conclusions

26-Thiodiosgenin (**3**) is readily available from diosgenin acetate by treatment with hydrogen sulfide/BF_3_·Et_2_O followed by hydrolysis [10]. The chemical reactivity of this compound has been investigated. We found that the MCPBA oxidation of 26-thiodiosgenin (**3**) can be performed chemo- and stereoselectively at −78 °C. Under these conditions, the C5-C6 double bond is not affected and the axial (*S*)-sulfoxide **6** is formed in a large excess: diastereomeric excess (*de*) around 75–80%. With 2.2 equiv. of MCPBA at −40 °C the major reaction product is sulfone **8**. The pure epimeric sulfoxides **6** and **7** rapidly isomerize in solution to afford an equilibrium mixture in the ratio of 3:2. The sulfoxide **6** and the sulfone **8** undergo deprotonation with n-BuLi and then can be alkylated with methyl or ethyl iodides. 26-Thiodiosgenin (**3**) has a strong antimicrobial activity against Gram-positive and Gram-negative bacteria. However, the *S*-oxidation and further α-methylation of **3** decreases its antimicrobial potential. The sulfoxides **7** and **10**, as well as alkylated sulfones **11** and **12**, exhibit a weak cytotoxicity against HeLa cell lines.

## Data Availability

Not applicable.

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
