# Peer review of "A Study on the Chemistry and Biological Activity of 26-Sulfur Analogs of Diosgenin: Synthesis of 26-Thiodiosgenin S-Mono- and Dioxides, and Their Alkyl Derivatives"

_molecules, 2022, doi:10.3390/molecules28010189_

Round 1
Reviewer 1 Report
The study has been carefully performed and clearly presented. Minor edits are suggested in the attached file.

Author Response
Thank you for your kind comments.
Line 27: combined is misspelled – corrected;
Line 46: nature not Nature – corrected;
Line 52: IC50 values are given without the units of concentration – concentration units have been added;
Line 75: Is compound referred to compound 3? – this is a general statement, but applies to compound 3 as well;
Line 77: reads better as … formation of a sulfoxide … - the indefinite article ‘a’ has been added;
Line 86,87: reads better as … the oxidation of organic sulfides … - changed as required;
Line 104: reads better as … protons of the C26 … - the definite article ‘the’ has been added;
Line 120: reads better as … was not possible … - the sentence is left unchanged so as not to change the meaning;
Line 133: Scheme 3, It would be helpful to label H-16a on the structures so a reader unfamiliar with steroid numbering does not have to refer to Figure 1 to identify the proton discussed – the carbon atom 16 has been labelled in Scheme 3;
Line 168 and Legend to Figure 4: presumable should be changed to presumed – changed to ‘tentative’;
Line 208: Suggested word change …synthesized by us … to … newly synthesized – changed as suggested;
Line 217: Delete – As it can be observed, - Start the sentence with - Diosgenin (1) – done as suggested;
Line 236: Change – not methylated to non-methylated – it has been changed;
Line 242,243: Change … allowed to conclude … to … allowed the conclusion that … - changed;
Line 263: Refering is misspelled – The error has been corrected.
Reviewer 2 Report
In the reviewed manuscript, the authors presented the synthesis of thioderivatives of diosgenin and their sulfone and sulfoxide analogs. In addition, the results of the antibacterial activity of the obtained compounds are presented in the manuscript. It is a pity that the authors limited themselves to only two strains of bacteria. However, the results of research on the interactions of the tested compounds with tryptophan were interesting, thanks to which the differences in the effects of the compounds on Gram (+) and Gram (-) bacteria can be partly explained. The cytotoxic activity of the compounds is negligible and frankly I do not know why it was studied at all. Cytotoxicity against normal human cell lines could be tested to verify that the compounds are not toxic. The whole part concerning the synthesis and explanation of the structure of the obtained derivatives does not raise objections and is very well written.
Author Response
It is a pity that the authors limited themselves to only two strains of bacteria - we used S. aureus (Gram-positive) and E. coli (Gram-negative) as the strains that are the most often used as standard, model bacterial cells in such studies. This allowed us to demonstrate the differences in the interaction of newly synthesized compounds with different types of bacteria.
The cytotoxic activity of the compounds is negligible and frankly I do not know why it was studied at all - the cytotoxicity of our compounds was studied because there were previous reports in the literature on the cytotoxicity of 26-thiodiosgenin against certain cell lines. The Reviewer is right that the cytotoxic activity of newly synthesized analogs is low. Usually, glycosides are much more active, but they have not been studied. On the other hand, in relation to normal cells (line RPE-1), the results are interesting, especially when compared with antibacterial activity against Staphylococcus aureus. A moderate cytotoxicity against HeLa cells was demonstrated for compounds 7, 10, 12, and the carbaanalog 5. This is important because the studied compounds can be modified to ultimately obtain a stronger anticancer activity.
Thank the Reviewer for his comments.
Reviewer 3 Report
Title: A Study on the Chemistry and Biological Activity of 26-Sulfur 2 Analogs of Diosgenin. Synthesis of 26-Thiodiosgenin S-Mono- 3 and Dioxides, and their Alkyl Derivatives
The manuscript is a good innovative work and presented sufficient evidence in support of their findings. However, some minor modification is required. A few facts are needed to be justified and some typographical and grammatical errors need to be corrected.
1. The author needs to mention the method opted for the synthesis of 26-thiodiosgenin in the main section, also, Line no 80 to 86 is not properly cited as per the mentioned facts.
2. The author needs to justify the reason to select antibacterial activity along with cytotoxicity assay for the synthesized derivatives.
3. Sulphur-containing antimicrobial drugs or drug candidates in preclinical studies should be incorporated into the manuscript and following reference can be cited
: Mini Rev Med Chem Anticancer Activity of Diosgenin and Its Semi-synthetic Derivatives: Role in Autophagy Mediated Cell Death and Induction of Apoptosis 2021;21(13):1646-1665..
Author Response
Thank you for your comments.
- The author needs to mention the method opted for the synthesis of 26-thiodiosgenin in the main section, also, line no 80-86 is not properly cited as per the mentioned facts – the following sentence has been added: ‘In particular, the one-step Wang synthesis involving the treatment of a solution of diosgenin in dichloromethane with hydrogen sulfide gas in the presence of BF3·Et2O as a catalyst, turned out to be the most advantageous and was chosen by us.’
- The authors needs to justify the reason to select antibacterial activity along with cytotoxicity assay for the synthesized derivatives – the reason was to study a wide spectrum of the biological potential of sulfur analogs of diosgenin. We have chosen for this study bacterial cells and human cancer cells to demonstrate biological activity of newly synthesized compounds against prokaryotic and eukaryotic cells. The studies were also extended by the use of RPE-1 cells in order to verify if these compounds are non-toxic to normal cells. This allowed us to get to know better their varied activity in biological systems at the cellular level.
- Sulphur-containing antimicrobial drugs or drug candidates in preclinical studies should be incorporated into the manuscript and following reference … can be cited – on page 9 of the original manuscript we wrote: ‘It is well known that sulfur functional groups are found in many pharmaceuticals, including penicillin, sulfamethoxazole, lansoprazole, or dapsone [23]’. In the revised manuscript we have slightly changed and extended the above phrase by adding some new drugs of different classes of organic, sulfur compounds. Furthermore, the suggested paper has been cited as ref. [11].